# A Simple Stochastic Reaction Model for Heterogeneous Polymerizations

**DOI:** 10.3390/polym14163269

**Published:** 2022-08-11

**Authors:** Jiashu Ma, Jiahao Li, Bingbing Yang, Siwen Liu, Bang-Ping Jiang, Shichen Ji, Xing-Can Shen

**Affiliations:** State Key Laboratory for Chemistry and Molecular Engineering of Medicinal Resources, Key Laboratory for Chemistry and Molecular Engineering of Medicinal Resources (Ministry of Education of China), Collaborative Innovation Center for Guangxi Ethnic Medicine, School of Chemistry and Pharmaceutical Sciences, Guangxi Normal University, Guilin 541004, China

**Keywords:** stochastic reaction model, heterogeneous polymerization, homogeneous polymerization, reaction microenvironment, Monte Carlo simulation

## Abstract

The stochastic reaction model (SRM) treats polymerization as a pure probability‐based issue, which is widely applied to simulate various polymerization processes. However, in many studies, active centers were assumed to react with the same probability, which cannot reflect the heterogeneous reaction microenvironment in heterogeneous polymerizations. Recently, we have proposed a simple SRM, in which the reaction probability of an active center is directly determined by the local reaction microenvironment. In this paper, we compared this simple SRM with other SRMs by examining living polymerizations with randomly dispersed and spatially localized initiators. The results confirmed that the reaction microenvironment plays an important role in heterogeneous polymerizations. This simple SRM provides a good choice to simulate various polymerizations.

## 1. Introduction

Heterogeneous polymerizations are polymerization processes with two or more phases during or after polymerization, as well as polymerizations at the surface or interface [1,2]. Typical heterogeneous polymerizations, such as suspension, emulsion, dispersion, precipitation polymerizations, etc., are widely applied for the synthesis of high-performance materials such as paints, coating, and many others. Other systems include polymerization-induced phase separation (PISA) [3,4,5], surface-initiated polymerization (SIP) [6,7], and polymerizations in living cells [8,9]. It is challenging to characterize the polymerization kinetics since the distributions of species are inhomogeneous in space and variant with time.

Today, computer simulation is an important tool in the study of polymerization, which can provide both microscopic structural and time-dependent information about the system. Due to the broad coverage of the time and length scales of polymerization systems, coarse-grained models are usually utilized to describe the polymerization kinetics, such as coarse-grained molecular dynamics simulation (CGMD) [10,11,12,13,14,15,16,17,18], Brownian dynamic simulation [19], the dynamic lattice liquid model (DLL) [20,21,22], the bond fluctuation model (BFM) [23,24,25,26,27], the reaction-event-driven kinetic Monte Carlo (*k*MC) model [28,29], and the dissipative particle dynamics (DPD) method [30,31,32,33,34,35], etc. Various strategies, based on the distance, probability, or reactive force field, have been applied to manipulate the polymerization [15,36].

We are interested in the stochastic reaction model (SRM) [37,38], which treats polymerization as a pure probability-based issue. Due to the simple idea, various SRMs have been proposed and applied in CG simulations to study polymerizations in homogeneous as well as heterogeneous systems [10,12,19,25,27,31,32,33,34,35]. However, as pointed out by Arraez et al., most SRMs are based on simplified constant reaction probabilities, which fail to capture the kinetics of heterogeneous polymerizations since the concentration variations are inevitable [29].

To better understand the dilemma of the SRM, we would like to briefly introduce how to simulate a living polymerization with different versions of the SRM. As known, the polymerization rate *R_p_* of a living polymerization is
(1)$$R_{p}=-\frac{d\left[M\right]}{dt}=k_{p}\left[M\right]{\left[I\right]}_{0},$$
where *k_p_* is the reaction rate constant, and [*M*] and [*I*]_0_ are the concentrations of free monomers and active centers, respectively. A successful SRM should yield the correct polymerization kinetics.

In the SRM, usually, an active center reacts with a randomly selected free monomer [27,33] or a closed one [39,40] in the given reaction radius with a probability *P_r_* (Figure 1a). The number of consumed monomers in one reaction step is *I*_0_*P_r_*. *P_r_* is an important characteristic for controlling the kinetics of the process. According to Equation (1), *P_r_* should be proportional to the concentration of free monomers. In a very popular version of the SRM (Version I), *P_r_* is simply calculated as [27]
(2)$$P_{r}={P_{0}\left[M\right]/\left[M\right]}_{0}$$
where [*M*] and [*M*]_0_ are the instantaneous and initial concentration of free monomers, respectively, and the initial addition probability *P*_0_ is a constant. Equation (2) implies that the system is well-mixed, i.e., all species are homogeneously distributed in space so that all active centers react with the same *P_r_* in Version I.

In Version I, the reaction probability *P_r_* decreases with the decreasing of the monomer concentration [*M*] and should be calculated at every reaction step. When the monomer conversion is low, the variance of [*M*] can be ignored [36]. There is no need to calculate *P_r_* at every reaction step. In some simulations (Version II), *P_r_* was further simplified as a constant, which was independent of [*M*] [31,33,41]. It is a type of polymerization with a constant reaction rate *R_p_* instead of a constant reaction rate constant *k_p_*.

Version I differs from Version II, as the influence of the monomer concentration has been considered in Version I. However, in both versions, all active centers react with the same *P_r_*. The divergence of the local reaction microenvironment is not considered. To simulate heterogeneous polymerizations, a step forward is required to consider the reaction microenvironment of each active center [29].

In the DPDChem software [42], another strategy has been applied. An active center sequentially reacts with nearby monomers according to the distance with a fixed probability of *P*_0_ until a bond is created or there are no more unchecked monomers (Figure 1b). We can calculate the probability of the active center to create a bond, which is
(3)$$P_{r}=1-{\left(1-P_{0}\right)}^{m},$$
where *m* is the number of monomers in the reaction radius. When *P*_0_ is small, *P_r_* can be expressed as *mP*_0_. In this version (Version III), each active center reacts with its *P_r_*, and the effect of the reaction microenvironment is well considered. DPDChem has been widely applied to simulate various homogeneous and heterogeneous polymerizations [35,43,44,45,46,47].

In lattice Monte Carlo (MC) simulations, such as BFM and DLL [20,21,22,23,24,25,26,48,49], another version (Version IV) has been applied (Figure 1c). For an active center, one of the neighboring sites is randomly selected. If this site is occupied by a free monomer, a reaction is tried with a fixed probability *P*_0_. Attention must be paid that Version IV (select a site and try to react if it is a monomer) is essentially different from Versions I and II (select a monomer and try to react). The reaction microenvironment is considered in Version IV since the probability to find a free monomer at a given site is determined by the local concentration of free monomers. On the contrary, it is not considered in Versions I and II.

Recently, we proposed a simple SRM (Version V) for heterogeneous polymerizations [38]. When an active center tries to react with a randomly selected free monomer in the reaction radius, the reaction probability *P_r_* is calculated according to the number of local free monomers *m* as *mP*_0_ (Figure 1a). This idea follows the work by Akkermans et al. [10], in which the probability was suggested to be *m*/*m*_max_, where *m*_max_ was a preset value and always larger than *m*. To our understanding, however, it was the average number of free monomers <*m*> that was applied in [10], which approached a constant value for not short chains (Equation (4) in [10]). We applied version V as well as Version I for comparison to study surface-initiated polymerization (SIP) [38]. Essentially, SIP is a heterogeneous polymerization that is easy to be ignored because free monomers are gradually distributed in the system [38]. The results suggested that the reaction microenvironment plays an important role in SIP.

To our understanding, the influence of the reaction microenvironment has not attracted enough attention in many simulations with SRM. For example, inappropriate versions have been applied in the simulations of SIP and PISA [27,32,33]. Sometimes, the description of the SRM is too simple to judge whether the heterogeneous reaction microenvironment has been considered or not. To provoke enough attention to the reaction microenvironment and promote the application of the SRM in heterogeneous polymerizations, we compared the new SRM with Versions I, II, and III in this paper. Version IV is not examined here as it can be only applied in lattice simulations. Theoretically, it should obtain the same results as Version V. The chance that a randomly selected neighbor site is occupied by a free monomer is proportional to the number of local free monomers. The algorithms of different versions of the SRM are introduced in Section 2. The results of living polymerizations with randomly dispersed and spatially localized initiators are shown in Section 3. A brief conclusion is given in Section 4.

## 2. Models and Simulation Methods

### 2.1. Lattice Monte Carlo Simulation

The simulation was carried out in a simple cubic lattice with a volume *V* = *L* × *L* × *L* with the periodic boundary condition in three directions. The Larson-type bond fluctuation model has been adopted [38,50,51,52] since the corresponding theoretical polymerization kinetics can be easily obtained to guide the simulation, as shown later. In this model, each monomer (or initiator) occupies one lattice, and the permitted bond length is 1 or √2. During relaxation, a monomer randomly selects one of 18 nearest and next nearest neighbor sites and tries to move. In any elementary movement, a bond intersection is forbidden. Meanwhile, each lattice can be occupied only once; thus, the excluded volume effect is well considered in this simulation. The stimulation time is measured in units of MC steps (MCs), which is defined as all monomers are tried to move once on average.

### 2.2. Implementation of Stochastic Reaction Model

While implementing SRM, one question is when to model a reaction. In the work by Genzer [27], a decision was made by a comparison of a generated random number with a probability of choosing motion over reaction. While studying the influence of diffusion with BFM, Lu and Ding applied a probability to reduce the movement of a monomer to a vacancy [24]. Here, a characteristic delay time, or reaction interval time, *τ* was applied to separate two successive reaction steps [33,38,53]. The effect of diffusion can be simply tuned by adjusting the value of *τ*.

When an active center tries to react with a free monomer in a given reaction radius *R_cut_*, the value of *R_cut_* influences the polymerization kinetics. A bigger *R_cut_* can speed up the polymerization, as there are more reaction candidates. In simulations such as MD, however, instabilities might be provoked due to the strong force after the creation of a new bond [15]. A proper reaction radius is needed to achieve the balance between the polymerization speed and the simulation stability. A larger reaction interval time is helpful for the dissipation of the new bond energy and the simulation stability. Here, *R_cut_* was set as √2, the same as the longest bond length in the Larson-type bond fluctuation model. The maximum reaction candidates *m*_max_ around an active center is 18.

Section 1 has introduced the basic idea of different SRMs. In this simulation, the procedures of different versions are as follows: In Version III, an active center sequentially reacted with nearby monomers with a reaction probability *P_r,seq_* = *P*_0_, where *P*_0_ is a constant and defined as the reaction probability between one active center and one free monomer. In Version V, an active center tried to react with a randomly selected free monomer in the reaction radius with a reaction probability *P_r,loc_* = *mP*_0_, where *m* is the number of monomers in the reaction radius of the active center. Here, *P*_0_ is restricted to be no larger than 1/*m*_max_, i.e., *P_r,loc_* is no larger than 1; thus, the effect of the concentration of free monomers could be correctly considered.

In the literature [27,36], the reaction probability of Version I was calculated according to the instantaneous concentration of free monomers [*M*] and scaled by the initial concentration [*M*]_0_ (Equation (2)). It must be pointed out that [*M*]_0_ in Equation (2) should be the maximum concentration of monomers in a bulk polymerization. Otherwise, it will fail to compare the polymerization kinetics of systems with different initial monomer concentrations. In this study, the reaction probability of Version I, *P_r,av_*, was calculated according to the average number of free monomers around an active center
(4)$$P_{r,av}=(m_{\max}-1){\left[M\right]}_{t}P_{0}.$$

Here, *m*_max_ − 1 instead of *m*_max_ is used as one nearby position is occupied by the previous monomer of the same chain. For Version II, the reaction probability *P_r,const_* was a constant, which was calculated according to the initial concentration of free monomers as
(5)$$P_{r,const}=(m_{\max}-1){\left[M\right]}_{0}P_{0}.$$

Once the reaction probability (*P_r,av_*, *P_r,const_*, *P*_0_, and *P_r,loc_* in Versions I, II, III, and V, respectively) between the reaction pair is determined, a random number is generated. If the randomly generated number is no larger than the reaction probability, the reaction is accepted, and the free monomer turns out to be the active center for future reaction. The information such as the chain length will be updated. The illustrations of one reaction cycle of different versions are shown in Figure 1.

### 2.3. Polymerization Kinetics

The polymerization kinetics of a living polymerization has been deduced to guide the simulation [38]. Since the time between two reaction steps is *τ*, the concentration change of free monomers during polymerization can be calculated as
(6)$$-\frac{d\left[M\right]}{dt}={\left[I\right]}_{0}P_{r}/\tau .$$

According to Equation (4), the above equation can be written as
(7)$$-\frac{d\left[M\right]}{dt}=(m_{\max}-1){\left[I\right]}_{0}{\left[M\right]}_{t}P_{0}/\tau .$$

The theoretical monomer conversion of a homogeneous living polymerization is
(8)$$C=1-\frac{{\left[M\right]}_{t}}{{\left[M\right]}_{0}}=1-\exp(-(m_{\max}-1){\left[I\right]}_{0}P_{0}t/\tau ).$$

## 3. Results and Discussion

### 3.1. Homogeneous Polymerization

Firstly, different versions of the SRM were applied to examine a living polymerization system with randomly dispersed free monomers and initiators. The length of the cubic lattice was *L* = 60, the initial monomer concentration was [*M*]_0_ = 0.4 monomer per lattice, the number of initiators was *I*_0_ = 1000, the reaction interval time *τ* was 10 MCs, the simulation time was 10^6^ MCs, and the reaction probability between one active center and one monomer was *P*_0_ = 0.001. In the reaction step, an initiator is randomly selected, then it tries to react with a nearby free monomer with a certain reaction probability. If the reaction is accepted, the initiator transfers the reactivity to the free monomer and itself becomes the first monomer of the polymer chain. Meanwhile, the newly reacted free monomer becomes an active center for future reactions. In this way, linear polymer chains are obtained. The results were averaged over 20 independent runs.

Living polymerization characters were obtained with all versions, and the results are almost identical (Figure 2). For example, the linear relationship between the number-average molecular weight *M*_n_ and the monomer conversion *C* was observed (Figure 2a). The dispersity (*Đ* = *M*_w_/*M*_n_) slightly increased with *C* at the start of the simulation and further decreased to reach a plateau (Figure 2b). Figure 2c shows the molecular weight distributions during the polymerization, which can be described by the Poisson distribution
(9)$$P(N)=\frac{M_{n}{}^{N}}{N!}\exp(-M_{n}).$$

We studied the number of unreacted initiators *I* during polymerization. It was predicted that *I* can be calculated as [33]
(10)$$I=I_{0}{\left(1-P_{r}\right)}^{t/\tau }$$
when *P_r_* is a constant. In the early stage of the polymerization, the monomer conversion is very low, e.g., about 3% when *t* = 4000 MCs. The influence of monomer concentration on the reaction probability can be ignored. Figure 2d shows that ln(*I*/*I*_0_) decays linearly, consistent with the theoretical prediction very well.

The differences between different versions can be observed when the monomer conversion *C* and reaction rate *R_p_* are concerned. As expected, the monomer conversion *C* of Version II increased linearly with time and quickly reached a plateau due to the constant reaction probability (Figure 3a). The results of other versions followed the polymerization kinetics of a living polymerization as predicted by Equation (8). However, a deviation could be clearly observed for Version I in the late stage of the simulation. The reason is that the well-mixed assumption adopted in Version I is no longer held. When the concentration of free monomers [*M*] is very low (*m*_max_[*M*] < 1), a monomer cannot be found in the reaction radius of an active center. Both the chance to find a free monomer and the reaction probability *P_r,av_* are proportional to [*M*]. The effect of monomer concentration was considered twice, and a slow polymerization is observed (also shown in Figure 3b). Due to the same reason, the polymerization rate of Version II was no more a constant at the late stage of polymerization (Figure 3b). Both Versions I and II should better study polymerization systems with a concentration of free monomers higher than 1/*m*_max_ [38]. There is no such limitation for Versions III and V, and the results are consistent with the theoretical predictions even when the concentration of free monomers is very low (Figure 3a,b).

### 3.2. Heterogeneous Polymerization

We examined a heterogeneous polymerization system with 1000 initiators spatially localized in a 10 × 10 × 10 cube at the beginning of the simulation (as illustrated by the inset in Figure 4a), which corresponds to an experiment with powered initiators. The initiators are allowed to diffuse apart during the polymerization. We can expect that the polymerization should be influenced by the localization of the initiators since the inner and outer initiators react differently.

When the same parameters as those in Figure 2 were applied (*P*_0_ = 0.001 and *τ* = 10 MCs), the obtained results are similar to those of the homogeneous systems (Appendix A). According to the snapshots (Appendix A), the localized initiators became quickly dispersed after several reaction steps due to the slow polymerization and small size of the initiators. The effect of the localization of the initiators was not obvious.

When the polymerization is very fast (*P*_0_ = 0.01), the influence of the localization of the initiators can be observed. All versions suggest that the polymerization of the heterogeneous system (Figure 4a) is slower than that of the corresponding homogeneous system (Figure 4b). It is easy to explain. Due to the large *P*_0_, the initiators reacted with free monomers before they diffused apart from each other. The outer initiators have more chance to react with free monomers, the inner ones are trapped. Due to the trapping effect, the initiators should react slower, and a broader molecular weight distribution should be observed. Such a slower conversion of initiators was observed with Versions III and V (Figure 4c). Meanwhile, Versions I and II obtained a conversion similar to the theoretical prediction. Figure 4e suggests that the molecular weight distributions of the heterogeneous polymerization system are very broad and cannot be described by the Poisson distribution. For comparison, the results of the homogeneous polymerization system still follow the theoretical predictions even when the polymerization is very fast (Figure 4d,f). It confirms that the location of the initiators influences the polymerization. Firstly, the access of free monomers to an inner initiator (active center) is limited as the area around the initiator (active center) is occupied by other initiators (active centers) and reacted monomers. Secondly, the delivery of free monomers might be blocked by outer initiators and active centers due to the large *P*_0_. As a result, inner initiators (active centers) might be in a starve state, i.e., they fail to find a free monomer in the reaction range. It is the reason that a broad distribution was still obtained with Versions I and II. Therefore, the distribution revealed by Versions III and V is more reliable since the reaction microenvironment is well considered.

### 3.3. A Further Comparison between Versions III and V

In Versions III and V, the influence of the reaction microenvironment was considered as the reaction probability of an active center is determined by the local reaction microenvironment. As shown, both versions can be applied to study both homogeneous and heterogeneous polymerizations, and the obtained results are almost identical. Strictly speaking, the consideration of reaction microenvironment is different in Versions III and V. It is directly considered in Version V as the reaction probability *P_r,loc_* = *mP*_0_, while indirectly considered in Version III according to Equation (3). One question is whether this difference can be ignored or not.

The difference between Versions III and V can be estimated by the reaction probability. According to Equation (3), the reaction probability of Version III, *P_r,seq_*, can be expressed as *mP*_0_ when *P*_0_ and *m* are small. Figure 5a shows *P_r,seq_* as a function of *P*_0_ while fixing *m* = *m*_max_. *P_r,seq_* gradually deviates from the theoretical reaction probability *P_th_* = *mP*_0_ with increasing *P*_0_. We calculated the relative difference between the reaction probability
(11)$$\Delta P/P_{th}=\left(P_{th}-P_{r,seq}\right)/P_{th}.$$

The relative difference is 8.1% when *P*_0_ = 0.01 and *m* = 18. For a given *P*_0_, the relative difference decreases with decreasing *m*. As the concentration of free monomers [*M*] is 0.4 in this study, the relative difference is only 3.1% at the start of the stimulation and decreases with polymerization (decreasing *m*). Thus, no significant difference between Versions III and V can be observed in Figure 4. When *P*_0_ is extremely large (*P*_0_ = 0.05 and [*M*] = 0.4), the relative difference reaches 14.2%. Figure 5b shows that the polymerization of Version III is slower than that of Version V. However, the difference between the molecular weight distributions is still ignorable (Appendix A).

According to the communication with Berezkin, the reaction probability in the simulation should be sufficiently small as the reaction probability in the real world is far smaller than that in simulations. Otherwise, the overall kinetics becomes unrealistic because it will be diffusion controlled. Typically, the reaction probability is 0.001, or even smaller in simulations. With such a small value, polymerization can be freely studied by either Version III or V.

## 4. Conclusions

Due to the simple idea, various stochastic reaction models were proposed to simulate different polymerization processes, which can be divided into two types according to the reaction microenvironment considered or not. In the first type (Versions I and II), all active centers react with the same probability, ignoring the divergence of the reaction microenvironment, in the second type (Versions III, IV, and V), each active center reacts individually according to the local reaction environment, considering the influence of reaction microenvironment.

In this paper, we applied different versions of the SRM to study a homogeneous polymerization system with randomly dispersed initiators and a heterogeneous polymerization system with spatially localized initiators. For the homogeneous polymerization system, all versions obtained typical characteristics of a living polymerization such as the linear increase in *M*_n_ with the monomer conversion *C* and the Poisson molecular weight distribution. For the heterogeneous polymerization system with spatially localized initiators, the expected behaviors, such as the slow conversion of initiators and broader molecular weight distribution, were observed with the second type of the SRM (Versions III and V). Though Versions I and II obtained broad MWDs when polymerization is very fast, the results are no more reliable. Similar conclusions were also obtained from the study of surface-initiated polymerization [38] that the properties such as MWD and dispersity are strongly influenced by the reaction microenvironment.

In a summary, caution must be paid while studying heterogeneous polymerization with a stochastic reaction model. Each active center should react with its own probability, which is determined by the active center’s local reaction microenvironment. As far as the three versions (III, IV, and V) are concerned, we recommend Version V due to its simplicity, popularity for both lattice and off-lattice simulations, and convenience for theoretical analysis.

## Figures and Tables

**Figure 1 polymers-14-03269-f001:**
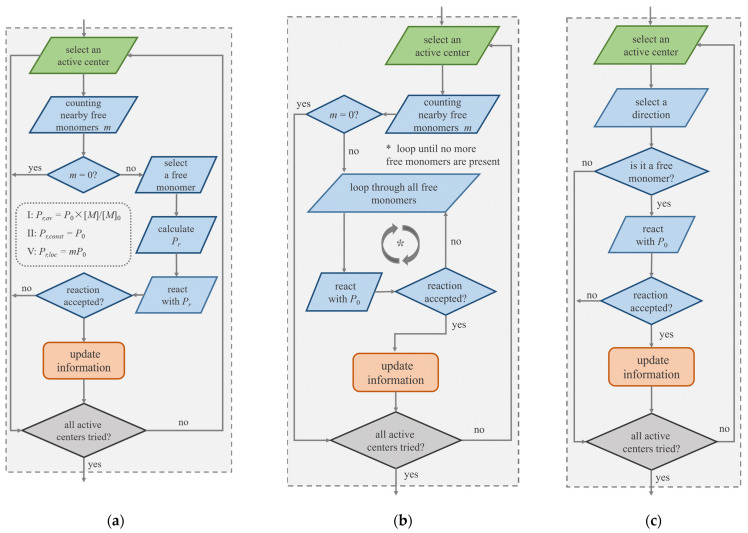
Illustration of one reactive cycle in different stochastic reaction models. (**a**) An active center reacts with a free monomer in the reaction radius with probability *P_r_*. The methods to calculate *P_r_* in Versions I, II, and V are shown. (**b**) In Version III, an active center sequentially reacts with nearby free monomers. (**c**) In Version IV, one neighboring direction of an active center is selected in a lattice simulation; if it is occupied by a monomer, a reaction is tried.

**Figure 2 polymers-14-03269-f002:**
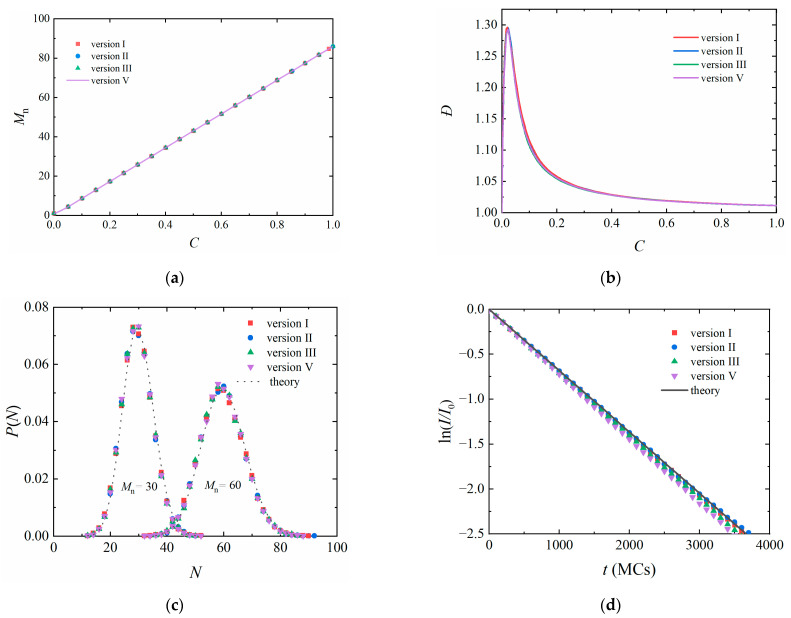
A living polymerization with randomly dispersed free monomers and initiators studied by different versions of stochastic reaction model. (**a**) Number-average molecular weight *M*_n_ and (**b**) dispersity (*Đ*) as a function of the monomer conversion *C*. (**c**) Molecular weight distribution with *M*_n_ = 30 and 60 respectively. The dotted lines are drawn according to a Poisson distribution. (**d**) The ratio of unreacted initiators during polymerization. The reaction interval time *τ* = 10 MCs, and the reaction probability between one active center and one monomer *P*_0_ = 0.001.

**Figure 3 polymers-14-03269-f003:**
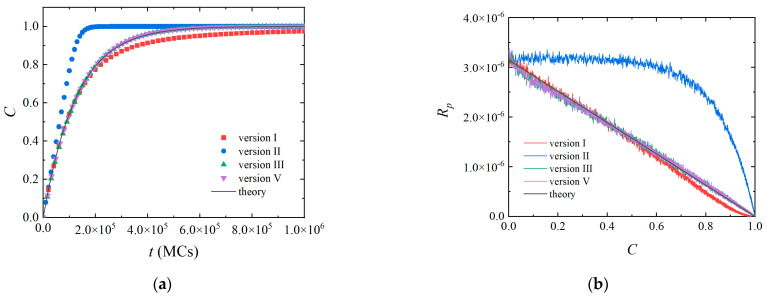
(**a**) Monomer conversion *C* as a function of simulation time. (**b**) Polymerization rate *R_p_* as a function of monomer conversion *C*. The parameters are the same as those in Figure 2.

**Figure 4 polymers-14-03269-f004:**
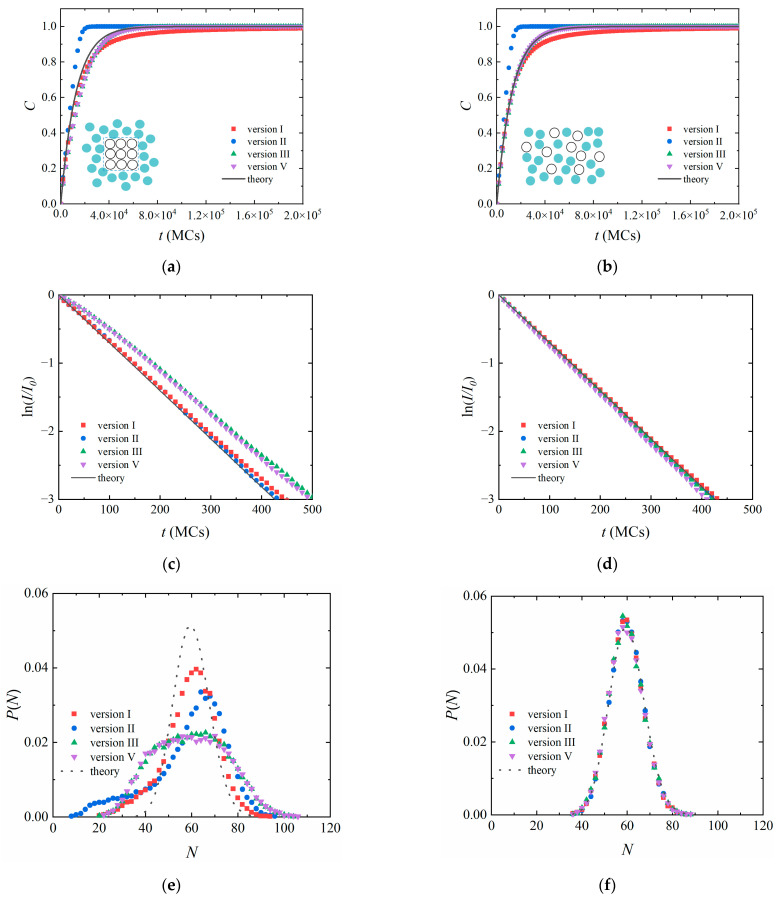
(**Left column**) Heterogeneous polymerization with spatially localized initiators; (**right column**) homogeneous polymerization with randomly dispersed initiators. (**a**,**b**) Monomer conversion *C*. (**c**,**d**) the ratio of unreacted initiators *I*/*I*_0_. (**e**,**f**) molecular weight distribution with *M*_n_ = 60. The dotted lines are Poisson distributions. The inlets are the 2D illustrations of the spatially localized and randomly distributed initiators, respectively. The cyan cycles: free monomers; hollow cycles: initiators. The number of initiators *I*_0_ = 1000, the reaction interval time *τ* = 10 MCs, and the reaction probability between one active center and one monomer *P*_0_ = 0.01.

**Figure 5 polymers-14-03269-f005:**
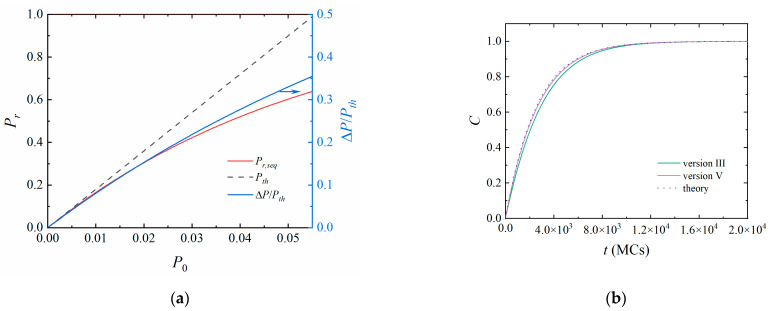
(**a**) The reaction probability *P_r,seq_*, *P_th_* and the relative difference as a function of *P*_0_. The number of monomers around each active center is *m* = 18. (**b**) The conversion of free monomers during the polymerization of a system with randomly distributed initiators. The reaction interval time *τ* = 10 MCs, and the reaction probability between one active center and one monomer *P*_0_ = 0.05. Other parameters are the same as those in Figure 2.

## Data Availability

The data that support the findings of this study are available from the corresponding authors upon reasonable request.

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
