# Peer review of "A Simple Stochastic Reaction Model for Heterogeneous Polymerizations"

_polymers, 2022, doi:10.3390/polym14163269_

Round 1

Reviewer 1 Report

The article by Jiashu Ma, Jiahao Li, Bingbing Yang, Siwen Liu, Bang-Ping Jiang, Shichen Ji, and Xing-Can Shen is devoted to comparing a version of the simple stochastic reaction model modified by authors with several versions of this model taken from the literature. This article is interesting from a methodical point of view and undoubtedly may be interesting to readers who are developing methods for simulating chemical reactions in silico. Before recommending this article to be published, it requires improvement to make it more understandable to readers. My comments are given below.

1. In Line 64, the authors write, “In some simulations (version II), Pr was further simplified as a constant to save simulation time [20,23,31].”

Here it should be clarified what probability and why remains constant? It looks like that, as monomers are consumed, [M]/[M0] will decrease and Pr will automatically decrease (which indirectly takes into account the microenvironment). For this reason, the reaction rate cannot remain constant. In the current version of the manuscript, the wording looks ambiguous. From the description made, no difference can be seen between versions I and II. The authors indirectly confirm this with the phrase in Line 68 (“In both versions I and II, all active centers react with the same Pr.”).

2. It should also be explained how the free monomer is selected in the neighborhood of the chosen one. Is this choice played as a random event, or is it realized from other considerations?

3. In Lines 87-88, the authors write, “The procedure is the same as those of versions I and II, but the reaction probability of an active center is calculated according to the number of local free monomers m.”

From this sentence, it is not entirely clear what is used (P0 or Pr) to decide whether a new bond has been created? As far as I am familiar with the publications about DPDChem, Pr is an auxiliary characteristic for controlling the kinetics of the process. If a pair of monomers is selected (located within the cutoff radius) according to Fig. 1b, then a random event is played. If the generated number is less than P0, then the link is formed. In this case, at each step of calling the procedure that implements the reaction, Pr will correspond to equation (3). What happens in the case of algorithms I, II, and V when deciding on the formation of a connection, Pr? The current explanation, it's hard to understand. The authors seem to explain this somewhat further down the text in section 2.2. But for ease of reading, it is better to do this when they are describing the algorithms I-V.

4. In Line132, the authors write, “But in simulations like MD, instabilities might be provoked due to the strong force after the creation of a new bond [27].”

It should be noted here that to eliminate the large forces that occur during bond formation in MD and DPD simulations, the polymerization reaction is invoked through N steps of integrating the equations of motion. In this case, N is chosen so that the emerging bonds and growing chains (subchains) have time to relax. This should be clarified for readers.

5. Eqs. 6-8 contain time. The authors use MC simulation, in which there is no time. For readers, it should be explained how the time step and micro (macro) MC steps are related to each other. Do the authors mean that t = number of MC steps? How true is this relationship?

 6. When the authors are describing the system under consideration in section 3.1, the following should be clarified. What is the valence of each monomer? What is the role of the initiator? Is it the monomer from which the chain begins to grow? Is the reactivity transferred to the attached monomer in this case (i.e., the initiating monomer loses its reactivity)? This should be explained.

8. Why did the authors not implement the IVth version of the polymerization algorithm?

9. At the beginning of section 3.2. Do I understand correctly that the authors mean by heterogeneous polymerization such initial conditions under which the initiator was localized in the center of the simulation cell (Fig. 4a)? At the same time, Fig. 4b corresponds to the case when all components were uniformly mixed, is that correct? This should be explained in words. It should also be explained how the distribution shown in Fig. 2 can be realized in a real experiment. 4a?

10. How were obtained the theoretical predictions shown in Fig. 4e?

Reviewer 2 Report

In the current report, the authors have compared four types of stochastic reaction model and demonstrated that the reaction probability with the consideration of the microenvironment would improve the applicability of the SRM. Thus, the current work provides useful insight among different SRM algorithms as benchmark, which would potentially attract some interests for the related researchers. Before it can be accepted for publishing on Polymers, the following questions need to be addressed.

Major points:

1.     As a stochastic reaction model, I think the authors were missing some important references from Prof. Xuehao He’s group (BFM lattice model and CG-RMD model) at Tianjin University and from Prof. Erik Nies’s group (CG-RMD model) at KU Leuven.

Prof. Xuehao He:

a.     RSC Adv., 2014,4, 56625-56636; https://doi.org/10.1039/C4RA10271A

b.     https://doi.org/10.1002/pola.26058

c.     J. Chem. Phys. 134, 104901 (2011); https://doi.org/10.1063/1.3560643

Prof. Erik Nies:

d.     Adv. Theory Simul., 2: 1800102 (2019). https://doi.org/10.1002/adts.201800102

e.     J. Comput. Chem. 2018, 39, 1764– 1778. DOI: 10.1002/jcc.25348

2.     The simulation details need to be improved with more details for data reproducibility, especially for section 2.1.

3.     What is the difference between the Larson-type model and the 3DBFM? Are they actually the same method? The authors have mentioned that the Version IV algorithm is for BFM and DLL models, but why in the results and discussions there is no results presented for Version IV algorithm?

4.     For the heterogeneous polymerizations, the authors demonstrated that the polymerization could be freely studied by either version III or V if the reaction probability is small enough. Isn’t it a diffusion related issue? With smaller reaction probability, the chance for forming a new bond is low and the unreacted monomers have more time to diffuse. Could the authors provide some evidence for the heterogeneity? For example, the static structure factor. I think the heterogeneity for large reaction probability with be higher than that of the system with small reaction probability.

Minor points:

a.     The email address of Xing-Can Shen in the main paper seems to be not consistent with the one provided in the supporting information, the authors may want to double check it.

b.     For the second paragraph of the introduction, the authors should not pack all related reference together, i.e., reference [10-25], instead, the corresponding reference should appear after particular methods.

c.     Line 51: “the polymerization rate of Rp a living polymerization is” should be “the polymerization rate Rpof a living polymerization is”.

Round 2

Reviewer 1 Report

The authors answered all my questions and revised their manuscript. In the current version, it can be recommended for publication.